# Restorative Environmental Perception's Influence on Post-Tour Behavior of Desert Off-Road Self-Driving Tourists: The Mediating Role of Flow Experience

**Cai Li** [1,2,*] **and Jingyi He** [1,2]

[1]   Key Laboratory of the Sustainable Development of Xinjiang's Historical and Cultural Tourism, Xinjiang University, Urumqi 830046, China; hjyjyo@163.com
[2]   College of Tourism, Xinjiang University, Urumqi 830049, China
*   Correspondence: llcc@xju.edu.cn; Tel.: +86-13199850060

**Abstract:** Desert tourism has always been one of the main types of adventure tourism in the world. Restorative environmental perception and flow experience are closely related concepts in adventure activities, but few studies have examined the mechanisms of their effects on tourists' post-tour behaviors. Desert off-road self-driving tours, as a typical adventure tourism project, are suitable for studying the relationships among three factors. We chose the Kumutag Desert as our study area, as it attracts many off-road self-driving tourists every year. Based on SOR theory, data were collected through questionnaires, and partial least squares structural equation models were constructed using SmartPLS 3.0 software. It was found that (1) restorative environmental perception has a direct positive impact on visitors' flow experiences; (2) flow experience has a direct positive influence on tourists' revisit behaviors and recommendation behaviors; (3) novelty, escape, fascination, and compatibility in restorative environmental perception have a direct positive influence on tourists' post-tour behaviors; (4) flow experience partially mediates the relationships among novelty, escape, fascination, compatibility, and post-tour behavior. This article deepens the understanding of the mechanism by which restorative environmental perception and flow experience affect tourists' post-tour behaviors while expanding the scope through three studies and providing theoretical support for subsequent research on adventure tourism.

**Keywords:** restorative environmental perception; flow experience; post-tour behavior; sor model; adventure tourism

## 1. Introduction

Due to the accelerated pace of life, the stress and fatigue that modern people endure daily is growing. Coupled with the outbreak of global crises such as disease and economic recession, society is likely to face an increasing number of health challenges and behavioral disorders. To recover the mental, physical, and social capacities that are constantly depleted in life and work, people need to engage in restorative activities. Rest and relaxation are by far the most basic motivations for people to participate in leisure tourism, and the stunning natural scenery of destinations enables people to relax and relieves their fatigue [1]. Restorative environments provide restorative experiences and allow people to engage in restorative activities, i.e., restorative environments can help people recover from the negative emotions that accompany mental fatigue and stress [2]. The number of travelers visiting destinations for the purpose of relaxation, exploration, and wonder is growing, and this travel consumption trend represents a shift in travel demand from traditional sightseeing vacations to more challenging trips for spiritual recovery and the pursuit of self-worth. Since the 1980s, desert tourism has gradually changed from an explorer's paradise to a public tourism project due to its unique charm and has become popular among tourists, becoming one of the five fashionable types of tourism in the

21st century [3]. With the rise of desert tourism, academics have also conducted in-depth research in fields related to desert tourism. The narrow definition of desert tourism shows that the travel goal of tourists is to seek a hunting and exploratory travel experience, desiring to escape from reality and seek their true selves through desert travel [4]. The main modes of transportation taken by desert tourists are hiking and off-road self-driving, which are also the main types of activities for desert tourists [5]. Currently, the majority of desert tourists choose to travel by car, with the number increasing annually. Compared with sightseeing tours, which mainly focus on natural scenery, theme parks, and cultural relics, desert off-road self-driving tours are a type of adventure tourism which cannot only meet people's demand for entertainment and stress relief but also meet the special needs of tourists such as adventure, hunting, and seeking excitement. Desert off-road self-driving tourists can overcome difficulties through the exercise of skills and gain a sense of fun and achievement from the trip, making the desert off-road self-driving tour a kind of enjoyment trip to overcome nature and oneself. Xinjiang is the richest region of desert resources in China, among which the Kumutag Desert has very little rock and vegetation cover, with sand dunes, sand nests, Gobi, and other landforms, so it is much liked by global autocross enthusiasts. We chose the Longmen Inn off-road base at the northern edge of the Kumutag Desert as the field research site, which is the starting point for off-road self-driving tourists to explore the desert.

The feelings of engagement, forgetfulness, and excitement that arise during adventure tourism are often associated with flow experience, which is an optimal psychological state in which individuals are very engaged in the activity and experience a high level of pleasure [6]. Adventure tourists seek self-fulfillment and a sense of detachment from reality, but few studies have explored the post-tour behaviors of adventure tourists to provide follow-up support for the 'highest experience'. Desert off-road self-driving visitors enter the desert first, perceive the restorative environment, and then participate in driving activities to generate a flow experience, yielding an excellent travel experience. Restorative environmental perception affects visitors' flow experiences [7] and post-tour behaviors [1], but it is unclear how restorative environmental perception affects visitors' flow experiences and post-tour behaviors during desert off-road self-driving tours. Cater [8] pointed out that people enjoy the flow experiences generated during adventure activities and want to stay involved in the activity for a long time to experience the wonderful feeling again, so the flow experience will undoubtedly have an impact on tourists' post-tour behaviors. The post-tour behaviors of tourists are a key element of tourism research, but the mechanisms of the effects of restorative environmental perception and flow experience on the post-tour behaviors of desert off-road self-driving tourists are not deeply understood. Therefore, we will explore the relationships among restorative environmental perception, flow experience, and tourists' post-tour behaviors during desert off-road self-driving tours. Structural equation modeling based on SOR theory was used to explore the mechanism of action among the three factors and whether flow experiences act as a mediator. This study theoretically advances the research related to restorative environmental perception and flow experience, broadens the understanding of the influence of tourists' post-tour behaviors, and provides a theoretical perspective for improving the quality of desert off-road self-driving tourists' tours.

## 2. Literature Review and Research Hypothesis

### 2.1. SOR Model

The stimulus–organism–response (SOR) model proposed by Mehrabian and Rusell is one of the foundational models of environmental psychology and consists of three components: environmental stimuli, emotional state, and tendency behavior [9]. Stimuli from the environment can affect the physiological and psychological states of visitors; pleasure and arousal are common emotional states resulting from environmental stimuli; tendency behaviors impact visitors' post-tour behaviors, including the positive intention to return to the environment and the negative intention to leave it.

The SOR model is often used in tourism research to explore the mechanisms by which variables influence tourist behavior. Song et al. [10] explored whether the efficiency of social network interaction can motivate tourists' product purchase behaviors through the SOR framework and found that online interactions generate tourism purchase behaviors through the mediating effect of immersion and perceived value. Based on the SOR theory, Chen et al. [11] explored the mechanism of the influence of the tourism promo genre on tourists' behavioral intentions in crisis situations and proved that the emotional image of the destination fully mediates the relationship between the two. Based on the SOR theoretical model, Geng and Li [12] took travel apps as the research object to investigate the influence of "relationship embedding" on "purchase intention" and test the mediating effect of perceived value. Based on the above findings, the SOR model can be used to explain the effects of environmental stimuli on tourists' tendency behaviors and the mediating role of emotional state.

Huang et al. [13] applied the SOR model to attentional restoration theory to explore the influence of tourists' restorative perceptions on their post-tour behavioral intentions. A model based on the SOR structure was proposed and tested by Yang et al. [14]. The results show that the flow experience conveyed by virtual tourism experiences affects visitors' propensities to use virtual tourism and their willingness to spend. Based on the SOR model, Chen et al. [15] proposed a chain mediation model to reveal the influence of the wellness tourism experience landscape on tourists' revisit behaviors. The above studies show that restorative environmental perception, flow experience, and post-tour behaviors are suitable as research variables for the SOR model, but few studies have explored the relationships among the three using the SOR theory. Therefore, we use the SOR model to study desert off-road self-driving tours to construct a model of the relationships among tourists' restorative environmental perceptions (environmental stimuli), flow experiences (emotional state), and post-tour behaviors (tendency behaviors), and clarify the mechanism of interaction among the three.

## 2.2. Restorative Environmental Perception

The concept of restorative environmental perception is based on an extension of attention restoration theory (ART) proposed by Kaplan et al. ART theory suggests that people must focus their attention on accomplishing work and tasks in their lives, and this process evokes directed attention mechanisms. Continuous use of directed attention can easily lead to fatigue, and being in a restorative environment can help people relieve stress and physical and mental fatigue [16]. Restorative environments should have four qualities: being away, extent, fascination, and compatibility [17]. Fascination occurs when the environment is highly attractive and does not require people's deliberate, focused attention; being away occurs when people leave their habitual life environments to reduce their psychological and physical exhaustion; extent is the richness and continuity of the content of the environment to allow people to carry out relevant activities; compatibility occurs when the environment can provide activities that match the preferences and skill levels of visitors.

There are three main research hotspots in restorative environmental perception. The first is studying the restorative benefits of environmental types. Restorative measurements have been used in sites such as zoos [18], shopping centers [19], and natural environments [20]. Natural scenes dominated by greenery reduce stress, improve mood and concentration, and lower blood pressure [21]. "Blue space"-type environments (e.g., waterfronts and riverfronts) are vulnerable to changes in many environmental factors. However, "blue spaces" are more conducive to physical and mental recovery than "green spaces" (e.g., urban parks) [22]. Few studies have measured and analyzed restorative environmental perceptions of deserts, and it is not known whether restorative perceptions in desert environments differ from those in "green space" and "blue space". The second hot spot is scale development for restorative environmental perception. Based on Kaplan's [17] study, Hartig et al. [23] developed the Perceived Restorativeness Scale (PRS), which includes four

dimensions: fascination, compatibility, extent, and being away. Laumann et al. [24] further subdivided being away into physically away (novelty) and mentally away (escape) and developed the Restorative Components Scale (RCS), which consists of fascination, escape, novelty, extent, and compatibility. Pals et al. [18] created the Perceived Restorative Characteristics Questionnaire (PRCQ), which narrowed the definition of extent to coherence, a dimensional change that has since been found to be reliable in empirical studies. Lehto [25] invented the Perceived Destination Restorative Qualities Scale (PDRQS), which consists of the dimensions of fascination, extent, novelty, compatibility, escape, and discord. Based on the characteristics of Chinese culture, the discord dimension failed the reliability test and was not adopted in the study of Chinese tourists' restorative perceptions [26]. Using a sample of Jiuzhaigou tourists, Guo et al. [27] validated that the Restorative Environment Perception Scale, which consists of five dimensions: coherence, novelty, fascination, escape, and compatibility. It has been shown to have good reliability and validity. The scale has general applicability to Chinese tourists, and we designed the questionnaire with reference to the scale. The third hot spot involves examining the antecedents and consequences of the role of restorative environmental perception. Examples include the effects of environmental familiarity [28] and environmental preferences [29] on restorative environmental perception and the role of restorative environmental perception in regulating emotional well-being [30]. Most modern people use tourism to acquire restorative environmental perceptions, and tourism helps individuals improve their cognitive flexibility, stimulates creativity, and enhances physical and mental health [31]. Physical and mental health affects tourists' satisfaction and loyalty, which in turn affects their post-tour behaviors [32]. Therefore, the effect of restorative environmental perception on tourists' psychological states and the mechanism of restorative environmental perception on tourists' post-tour behaviors need to be explored in depth.

### 2.3. Flow Experience

Flow experience was proposed by psychologist Csikszentmihalyi [33], who considered flow experience to be a mental state in which the individual is fully immersed in the activity performed in the present moment, automatically filtering out irrelevant perceptions. It is an optimal experience that stimulates amazing creativity in individuals. Scholars focused on the composition characteristics and scale development of the flow experience in the early days, and the most widely used scale in flow experience research is the Flow Experience State Scale developed by Jackson and Marsh [34]. This scale contains nine dimensions; it has been tested to have good reliability and validity and has been recognized by researchers as a basic scale for subsequent studies of the flow experience. Based on previous studies, scholars have begun to conduct research on various aspects of flow experiences involving outdoor recreational activities [35], adventure recreation programs [8], mountaineering activities [36], and virtual tourism [37].

In recent years, flow experience has become a topic of interest in tourism experience to interpret the positive emotions of tourists [38,39]. Activities that produce flow experiences require people to set goals, use skills, improve their attention, exert a sense of control, and become fully immersed in the activity [33], so flow experience research is mostly associated with physical activity and adventure tourism [40,41]. The flow experience is a completely enjoyable experience, which is a key element of the tour experience and an important motivational tool that leads to repeated participation in tourism activities [34], so it is crucial to study the mechanisms of the flow experience on tourists' post-tour behaviors. There are few empirical studies on flow experiences in China, and most are theoretical in nature. Research has focused on education [42], sports [43], and online marketing [44], among other fields. To further apply the concept of flow experience to tourism behavior research in the Chinese context, there is an urgent need to generalize the existing empirical studies of flow theory.

### 2.4. Post-Tour Behavior

Post-tour behaviors are exhibited by tourists after a trip based on the emotional attitudes they experienced during the trip. Scholars have measured post-tour behavior in various dimensions. From a one-dimensional perspective, Žabkar et al. [45] defined it as "tourist loyalty", and Zhang [46] defined it as "sharing behavior". From a two-dimensional perspective, scholars such as Han [47], Kim et al. [48], and O Leary and Deegan [49] classified post-tour behavioral intention as "revisiting and recommending". In contrast, Chen and Qu [50] analyzed post-tour behavior from the three-dimensional perspective of "similar, revisit and recommendation behavior".

Post-tour behavior is an important indicator of the positive state of tourists after a tour, and analyzing post-tour behavior not only can increase the economic benefits of destinations but also has a driving effect on improving the quality of tourists' tours. Li et al. [51] found that in tourism art performance activities, tourists' live experiences significantly and positively influence their post-tour recommendation behaviors and suggested that managers focus on storytelling and atmosphere creation during tours to enhance tourists' experiences. Using rural tourism in Turpan as an entry point, Ye et al. [52] constructed a structural equation model of tourism motivation, tourism involvement and post-tour behavior to provide theoretical support for improving tourists' willingness to revisit and recommend rural tourism. Based on the comprehensive research above, we find that revisiting and recommending are the core parts of tourists' post-tour behaviors, and we use these two dimensions to measure the post-tour behaviors of desert off-road self-driving tourists.

### 2.5. Research Hypothesis

2.5.1. Restorative Environmental Perception and Flow Experience

Csikszentmihalyi [53] argued that nature influences flow experience by providing people with certain challenges, so there is a positive connection between these two concepts. It has been shown that restorative environments can facilitate the generation of flow experiences because they are often preferred sites for recreational activities and directed attention recovery, so restorative environmental perception is a predictor of flow experience [17,23]. Studies on the flow experiences of climbers in the Austrian region have confirmed that the restorative environmental perceptions of climbers positively influence their flow experiences [7]. Frochot et al. [54] found that the beautiful scenery of a ski resort enhanced the flow experiences of skiers. The magnificent landscape of the desert not only allows visitors to relieve stress and relax but also provides a platform for desert off-road self-driving visitors to challenge nature and themselves so that they can have a flow experience. Based on the above analysis, we propose the following hypothesis.

**H1.** Environmental restorative perception positively affects flow experience during desert off-road self-driving tours.

2.5.2. Flow Experience and Post-Tour Behavior

Over the past two decades, flow experience has been introduced into tourism research and is seen as a fundamental concept for a deeper understanding of the tourism experience. Studies of recreational mountaineering activities have confirmed that flow experiences positively influence visitors' experiences, which in turn influences their post-tour behaviors [32]. In outdoor tourism research, capturing tourists' emotional responses (i.e., flow experiences) is considered one of the most effective ways to understand their revisit behaviors [55]. The flow experience generated when participating in events such as music festivals and folklore shows increases visitors' willingness to participate in similar events again and to share their experiences [56–58]. A study of the flow experiences of ski eco-tourists found that flow experiences directly and positively influence tourists' intentions to revisit and recommend [59]. The flow experiences tourists have during rafting promote their positive emotions and satisfaction, which in turn influences their recommendation and revisit behaviors [60]. As an adventure tourism activity, desert off-road self-driving tours bring

tourists rich and explicit perceptions that can positively influence their post-tour behaviors. Therefore, we propose the following hypothesis.

**H2.** Flow experience positively affects tourists' post-tour behaviors during desert off-road self-driving tours.

### 2.5.3. Restorative Environmental Perception and Post-Tour Behavior

Restorative environmental perception, which is one of the tourists' motivations to travel, is often associated with post-tour behavior, which is a trip outcome variable. Zhou and Ye [61] used restorative environmental perception as a mediating variable to explore the mechanism of action between tourism involvement and tourists' willingness to re-visit and found that restorative environmental perception plays multiple chain-mediating roles in the structure of both. Using the context of cultural heritage tourism sites, Korean researchers found that the compatibility dimension had a positive effect on tourists' revisit intentions [62]. In a study of tourists' perceptions of restorative environments in Guangdong's Nankunshan tourist resort, Chen and Xi [1] found that the fascination and compatibility dimensions significantly and positively influence tourists' post-tour behavioral intentions. In a study on the restorative environmental perception of tourists in Kanas, Huang et al. [13] found that the novelty and escape dimensions significantly and positively influence post-tour behavior. Environmental psychology suggests that the restorative perceptions of visitors vary with the type of environment [17] and that there is not a parallel structure between restorative environmental perception dimensions but a differentiated and structural one [63]. Previous literature suggests that not all dimensions impact post-tour behavior, thus necessitating individual and meticulous examination of each dimension to refine the effects of restorative environmental perception on tourists' behavior in desert off-road scenarios. This will also allow exploration of the dimensions that hold significance in these scenarios. Deserts possess restorative elements such as vast spatial structures and novel natural scenery, which are more immersive to tourists than traditional scenic spots, so the unique desert environment may have a different impact on tourists' post-tour behaviors. During the self-driving process, tourists temporarily escape worldly worries, integrate themselves into the environment, and gain a restorative experience. This experience can make tourists deepen their favorable feelings for and identification with the destination, which in turn can influence their post-tour behaviors. Therefore, we propose the following hypothesis.

**H3a.** Coherence positively affects tourists' post-tour behaviors during desert off-road self-driving tours.

**H3b.** Novelty positively affects tourists' post-tour behaviors during desert off-road self-driving tours.

**H3c.** Escape positively affects tourists' post-tour behaviors during desert off-road self-driving tours.

**H3d.** Fascination positively affects tourists' post-tour behaviors during desert off-road self-driving tours.

**H3e.** Compatibility positively affects tourists' post-tour behaviors during desert off-road self-driving tours.

### 2.5.4. The Mediating Effect of the Flow Experience

Flow experience is often used as a mediating variable in tourism research. Xu [64] verified that flow experience mediates the influence of tourism service value on tourist satisfaction and suggested that tourism practitioners should focus on the role of flow experience in influencing tourist satisfaction. Zhang et al. [65] explored the mediating role of flow experience in tourists' viewing of short videos, using Ding Zhen's popularity as an

example, and confirmed that flow experience increases tourists' willingness to travel. From an immanent perspective, the flow experience mediates the effect of social media travel sharing on travelers' impulsive travel intentions [66]. From the perspective of physical strength, challenge, and risk, desert off-road self-driving tours are undoubtedly an adventure activity. Many studies have found that participants can derive flow experiences from adventure activities [67]. The desert is typically a place of attention recovery and where adventure happens. Self-driving tourists need to use skills, improve their concentration, and exert a sense of control during desert off-road to become fully immersed in the activity and gain a flow experience that further influences their post-tour behaviors based on their restorative environmental perceptions. Therefore, we propose the following hypothesis.

**H4a.** Flow experience mediates the relationship between coherence and post-tour behavior during desert off-road self-driving tours.

**H4b.** Flow experience mediates the relationship between novelty and post-tour behavior during desert off-road self-driving tours.

**H4c.** Flow experience mediates the relationship between escape and post-tour behavior during desert off-road self-driving tours.

**H4d.** Flow experience mediates the relationship between fascination and post-tour behavior during desert off-road self-driving tours.

**H4e.** Flow experience mediates the relationship between compatibility and post-tour behavior during desert off-road self-driving tours.

The research hypothesis model diagram is shown in Figure 1.

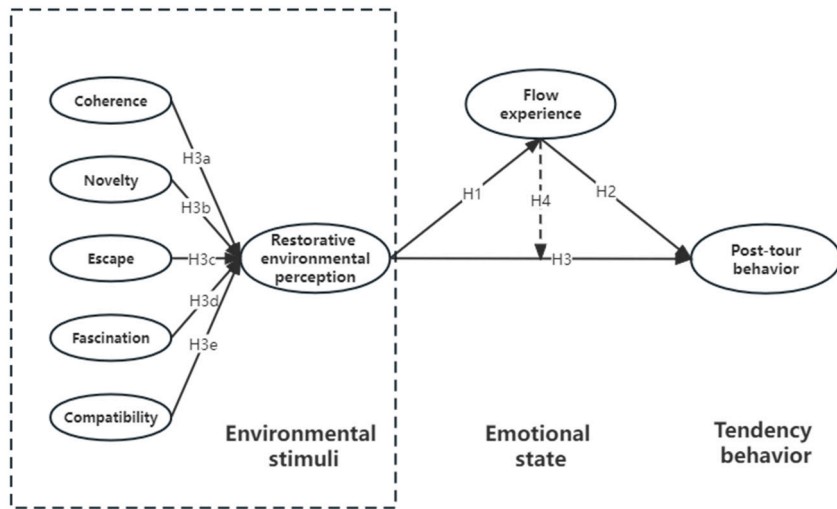

**Figure 1.** Theoretical model.

### 3. Methods

*3.1. Questionnaire Design*

The survey questionnaire consisted of four parts (Appendix A). The first section collected data on the demographic characteristics of the interviewed tourists. The first question was a discriminative item to determine whether the tourists had participated in desert off-road self-driving activities, and the remaining five items queried their gender, age, education, occupation, and annual frequency of desert off-road self-driving trips. The second section investigated the tourists' perceptions of the restorative environment using the five-dimensional structural scale of the restorative environment perception of Guo et al. [27]. The scale is generally applicable to Chinese tourists and has 21 question items, including five dimensions of coherence, novelty, escape, fascination, and compatibility. The third section investigated visitors' flow experiences using the Flow Experience Scale

developed by Jackson and Marsh [34]. This scale has shown good internal consistency and is widely used, with 27 question items across nine dimensions: a sense of time distortion, clear goals, unambiguous feedback, centering of attention, sense of control, loss of self-consciousness, integration of awareness and behavior, challenge-skill balance, and autotelic experience. The fourth section investigated the visitors' post-tour behaviors using two questions for the dimensions of recommendation and revisit. The above question items were revised for the desert scenario context and through preresearch interviews. Except for the first section, the remaining question items were closed-ended questions measured on a five-point Likert-type scale (1 = strongly disagree to 5 = strongly agree) to indicate the visitors' attitudes toward the question items.

### 3.2. Overview of the Study Area

The Kumutag Desert is located at the southern end of the old city of Shanshan County, Xinjiang, and is ranked as one of the eight deserts of China, with a total area of 2500 square kilometers. As the closest desert to the city, it has high accessibility and strong transportation convenience and is close to tourist cities such as Urumqi, Hami, and Korla, making its geographical location unique. This desert is in the temperate continental climate zone, with temperatures suitable for year-round adventure and recreation, sand sports, sand therapy, health care, sightseeing, and other activities. In 2015, the East Lake Auto Self-Driving Camp was built in the desert hinterland, providing a leisure place and living convenience for desert off-road self-driving tourists, explorers, and travelers. Compared with other deserts in China, the Kumutag Desert has high water content, fine and pure sand, and sand dunes with undulating drops of up to 100 m. The complete types of wind and sand landscapes and the peculiar landscape are conducive to the determination of the restorative environmental perceptions of tourists. The first new mode of tourism, combining city and desert, creates a sense of atmosphere far from the world and is vast and primitive, becoming a competitive paradise and beloved place for desert off-road self-driving enthusiasts.

The survey was distributed at the Dragon Inn off-road base, situated on the northern edge of the Kumutag Desert, spanning an area of 0.02 square kilometers. Proximity to the Kumutag Desert highway ensures convenient access for tourists. Additionally, a visitor service center is available, offering complimentary car and water refueling services. This base is a renowned attraction for self-driving enthusiasts who engage in desert off-road adventures. It serves as the starting point for desert off-road leaders such as the Bohai Off-road Club and the Avenue Off-road Club and serves as a cultural hub for off-road activities in Xinjiang. The research subjects here differ significantly from general tourists.

### 3.3. Data Collection

We used desert off-road self-driving tourists as the study population. Questionnaires were distributed through the Questionnaire Star platform and offline, and the research was divided into two stages. The first stage was the prestudy stage, conducted from 21 June 2022 to 27 June 2022 in an area with a high concentration of self-driving tourists at the Longmen Inn off-road base. The questionnaire was distributed by random interception. A total of 50 valid samples were collected, and the results of the analysis excluded the question items of the four dimensions with a Cronbach value below 0.7, and the reliability of the remaining dimensions was acceptable. Through interviews with professionals from the Xinjiang Bohai Off-road Club, Avenue Off-road Club, etc., the content of the questions was targeted and improved to be more conducive to understanding self-driving tourists, and a second stage of data collection was conducted. The second stage was the formal collection phase of the questionnaire, and an online questionnaire link was forwarded through WeChat, QQ and other social media channels. As of 6 August 2022, a total of 178 questionnaires were distributed, and 147 valid questionnaires were recovered after eliminating those with atypical completion times and those from duplicate IP addresses. The offline questionnaires were distributed at the Longmen Inn off-road base. A total of 270 questionnaires were distributed, and 257 valid questionnaires were collected after

eliminating invalid questionnaires. A total of 404 valid questionnaires were collected for an effective rate of 90%, resulting in a high recovery efficiency and good questionnaire quality.

## 4. Data Analysis

### 4.1. Sample Base Information

The demographic characteristics of the sample are shown in Table 1. In terms of gender, males accounted for 76.7% of the overall sample, and females accounted for 23.3%. The respondents were mainly between 21 and 50 years old, accounting for 85.1% of the total. In terms of education, 57.7% had a college education or above. In terms of occupation, 50.0% were business owners, 15.6% were corporate or government employees, and the rest worked in other industries. Of the sample, 70.8% visited the desert twice a year or more. The sample had good representativeness and enabled a follow-up study.

**Table 1.** Sociodemographic analysis of the sample.

| Variable | Categorization | Frequency | Percent |
|---|---|---|---|
| Gender | Male | 310 | 76.7 |
|  | Female | 94 | 23.3 |
| Age | 21–35 | 115 | 28.5 |
|  | 36–50 | 229 | 56.7 |
|  | 51–65 | 60 | 14.8 |
| Education | High School and below | 171 | 42.3 |
|  | Junior college | 111 | 27.5 |
|  | Bachelor's degree | 106 | 26.2 |
|  | Graduate degree and above | 16 | 4 |
| Occupation | Student or teacher | 16 | 4 |
|  | Business owner | 202 | 50 |
|  | Company employee | 38 | 9.4 |
|  | Government employees | 25 | 6.2 |
|  | Other | 123 | 30.4 |
| Annual frequency of desert off-road self-driving trips | Two times or less | 118 | 29.2 |
|  | Three to five times | 116 | 28.7 |
|  | Six to eight times | 46 | 11.4 |
|  | Eight times and above | 124 | 30.7 |

### 4.2. Reliability Testing

In this paper, we constructed a structural equation model with partial least squares (PLS) as the base path for subsequent empirical analysis. The gender of desert off-road self-driving tourists was predominantly male, and the data from the tourist portrait of local travel agencies in Shanshan County show that the ratio of male to female desert off-road self-driving tourists participating in the tour was approximately 7:3. Based on the results of fieldwork and interviews, it was clear that the proportion of male desert off-road self-driving tourists was higher. Therefore, we considered the gender ratio to be realistic. The partial least squares method is suitable for complex models and models with relatively small sample sizes and is suitable for developing theory because it does not require latent or significant variables to obey strict normal distribution assumptions to evaluate models [68]. Therefore, we chose SmartPLS 3.0 software to construct the structural equation model with partial least squares for the subsequent empirical analysis.

The results of the reliability tests are shown in Table 2. The Cronbach coefficient of each variable is greater than 0.8, which means that the reliability of each variable meets the requirements. The content of the measurement questions in the study was taken from the established scales. It was also pretested and revised by experts to ensure high content validity. Additionally, the standardized factor loading coefficients for each question item were greater than 0.7, the combined reliability (CR) for each variable was higher than 0.8, and the average extracted variance (AVE) for each variable was greater than 0.6, meaning

that all variables had good convergent validity [69]. According to the data in Table 3, it is clear that the arithmetic square root of the first-order variable AVE is greater than the Pearson correlation coefficient between the variables, indicating that the differential validity of the measurement model has passed the test [70].

**Table 2.** Factor structural validity model.

| Variables | Items | Factor Loading | First-Order Variable Factor Loadings | Cronbach's Alpha | CR | AVE |
|---|---|---|---|---|---|---|
| Fascination | FA1 | 0.803 | 0.91 | 0.89 | 0.93 | 0.76 |
| | FA2 | 0.872 | | | | |
| | FA3 | 0.917 | | | | |
| | FA4 | 0.892 | | | | |
| Escape | ES1 | 0.852 | 0.911 | 0.87 | 0.91 | 0.72 |
| | ES2 | 0.849 | | | | |
| | ES3 | 0.88 | | | | |
| | ES4 | 0.823 | | | | |
| Novelty | NO1 | 0.758 | 0.904 | 0.86 | 0.91 | 0.71 |
| | NO2 | 0.735 | | | | |
| | NO3 | 0.928 | | | | |
| | NO4 | 0.931 | | | | |
| Compatibility | CO1 | 0.868 | 0.909 | 0.92 | 0.94 | 0.75 |
| | CO2 | 0.859 | | | | |
| | CO3 | 0.878 | | | | |
| | CO4 | 0.852 | | | | |
| | CO5 | 0.886 | | | | |
| Challenge–skill Balance | CSB1 | 0.952 | 0.934 | 0.93 | 0.96 | 0.88 |
| | CSB2 | 0.933 | | | | |
| | CSB3 | 0.934 | | | | |
| Centering of Attention | COA1 | 0.921 | 0.914 | 0.91 | 0.94 | 0.85 |
| | COA2 | 0.924 | | | | |
| | COA3 | 0.914 | | | | |
| Unambiguous Feedback | UF1 | 0.939 | 0.94 | 0.93 | 0.96 | 0.88 |
| | UF2 | 0.936 | | | | |
| | UF3 | 0.934 | | | | |
| Clear Goals | CG1 | 0.912 | 0.941 | 0.9 | 0.94 | 0.83 |
| | CG2 | 0.937 | | | | |
| | CG3 | 0.887 | | | | |
| Sense of Control | SOC1 | 0.954 | 0.92 | 0.96 | 0.97 | 0.92 |
| | SOC2 | 0.966 | | | | |
| | SOC3 | 0.959 | | | | |
| Loss of Self-consciousness | LOSC1 | 0.846 | 0.919 | 0.85 | 0.91 | 0.77 |
| | LOSC2 | 0.919 | | | | |
| | LOSC3 | 0.864 | | | | |
| Post-tour Behavior | TB1 | 0.971 | | 0.94 | 0.97 | 0.94 |
| | TB2 | 0.97 | | | | |
| Restorative Environmental Perception | | | | 0.93 | 0.95 | 0.83 |
| Flow Experience | | | | 0.97 | 0.97 | 0.86 |

Note: FA = Fascination; ES = Escape; NO = Novelty; CO = Compatibility; CSB = Challenge–skill Balance; COA = Centering of Attention; UF = Unambiguous Feedback; CG = Clear Goals; SOC = Sense of Control; LOSC = Loss of Self-consciousness; TB = Post-tour Behavior.

### 4.3. Model Testing and Path Analysis

First, PLS calculations were performed using SmartPLS 3.0 software. The predictive explanatory power of the model constructs can be measured by $R^2$ values, with $R^2$ values of approximately 0.19, 0.33, and 0.67 representing weaker, moderate, and higher predictive explanatory power, respectively [71]. In this paper, restorative environmental perception and flow experience are conveyed by second-order reflective models. To ensure the accu-

racy of the models, the second-order reflective models were estimated using the average of the first-order model question item measurements. The $R^2$ values for the dimensions of unambiguous feedback, skill-challenge balance, sense of control, centering of attention, clear goals, and loss of self-consciousness were 0.88, 0.87, 0.85, 0.84, 0.89, and 0.85, respectively, all of which were greater than 0.67, indicating strong explanatory strength of the second-order constructs of the flow experience. The $R^2$ values for the dimensions of compatibility, novelty, escape, and fascination were 0.83, 0.82, 0.83, and 0.84, respectively, all of which were greater than 0.67, indicating that the second-order constructs of restorative environmental perception have strong explanatory strength. The $R^2$ value of the overall model is 0.60, indicating that it explains 60% of the variance in tourists' post-tour behaviors, indicating that the explanatory power of restorative environmental perception and flow experience in predicting tourists' post-tour behaviors is above moderate. Second, the $Q^2$ values obtained by the Blindfolding algorithm can measure the predictive relevance of the endogenous constructs. The $Q^2$ values for both flow experience ($Q^2$ = 0.59) and post-tour behavior ($Q^2$ = 0.55) were greater than 0.35, reaching a threshold value with strong predictive relevance [72]. Finally, since the evaluation of the partial least squares structural equation model is different from the traditional structural equation, the goodness-of-fit (GoF) index was calculated as a criterion for judging the goodness of fit of the model according to the suggestion of Unwin et al. [73], and a cutoff point of 0.10 for low, 0.25 for moderate, and 0.36 for high was used for the judgment according to the calculation of Wetzels et al. [74]. The GoF value of the model in this study is 0.73, which indicates that the overall goodness of fit of the model is high.

**Table 3.** Discriminant validity test.

| Variables | 1 | 2 | 3 | 4 | 5 | 6 | 7 | 8 | 9 | 10 | 11 |
|---|---|---|---|---|---|---|---|---|---|---|---|
| Compatibility | **0.87** | | | | | | | | | | |
| Unambiguous Feedback | 0.81 | **0.94** | | | | | | | | | |
| Challenge–skill Balance | 0.84 | 0.85 | **0.94** | | | | | | | | |
| Sense of Control | 0.76 | 0.83 | 0.87 | **0.96** | | | | | | | |
| Novelty | 0.76 | 0.67 | 0.58 | 0.54 | **0.84** | | | | | | |
| Centering of Attention | 0.83 | 0.82 | 0.85 | 0.78 | 0.62 | **0.92** | | | | | |
| Clear Goals | 0.79 | 0.89 | 0.84 | 0.84 | 0.64 | 0.82 | **0.91** | | | | |
| Post-tour Behavior | 0.69 | 0.7 | 0.64 | 0.57 | 0.64 | 0.73 | 0.76 | **0.97** | | | |
| Loss of Self-consciousness | 0.79 | 0.85 | 0.8 | 0.81 | 0.66 | 0.83 | 0.85 | 0.75 | **0.88** | | |
| Escape | 0.77 | 0.72 | 0.61 | 0.6 | 0.78 | 0.72 | 0.7 | 0.69 | 0.68 | **0.85** | |
| Fascination | 0.78 | 0.7 | 0.63 | 0.64 | 0.79 | 0.69 | 0.74 | 0.68 | 0.71 | 0.78 | **0.87** |

Note: The diagonal bolded values represent the square root of the first-order variable AVE values.

The bootstrapping program was used to analyze the path relationships between individual variables. The significance test needs to meet the criteria of a *t*-statistic greater than 1.96 and a *p*-value less than 0.05. First, H1 and H2 were tested. According to Table 4, the coefficients of the paths between the variables pass the significance test. It was found that (1) restorative environmental perception significantly and positively influenced flow experience and (2) flow experience significantly and positively influenced post-tour behavior, so hypotheses H1 and H2 were supported. Next, H3 was tested, and the results are shown in Table 4: the novelty, escape, fascination, and compatibility dimensions significantly and positively affected post-tour behavior, and H3b, H3c, H3d, and H3e were supported. At the same time, a preliminary judgment was made that there might be a mediating effect of flow experience in the relationships among novelty, escape, fascination, compatibility, and post-tour behavior, and the mediating effect was verified and analyzed below.

**Table 4.** Test results of the model.

| Path | Path Coefficient | *t*-Value | *p*-Value | Results |
|---|---|---|---|---|
| REP -> FE | 0.826 | 39.9 | 0 | support |
| FE -> PTB | 0.419 | 6.59 | 0 | support |
| ES -> PTB | 0.329 | 5.13 | 0 | support |
| NO -> PTB | 0.263 | 4.29 | 0 | support |
| FA -> PTB | 0.284 | 4.448 | 0 | support |
| CO -> PTB | 0.2 | 2.235 | 0.025 | support |

Note: REP = Restorative Environmental Perception; FE = Flow Experience; PTB = Post-tour Behavior; ES = Escape; NO = Novelty; FA = Fascination; CO = Compatibility.

*4.4. Intermediation Effect Test*

The bootstrapping procedure was again applied to verify the mediating effect of the flow experience. The bootstrapping procedure was set up to repeat the sampling 5000 times, and the results of the significance test for mediating effects are shown in Table 5. The mediation effect of the path is significant if the upper and lower limits of the 95% confidence interval do not contain 0. As shown in Table 5, the flow experience mediates the relationships among novelty, escape, fascination, compatibility, and post-tour behavior, and H4b, H4c, H4d, and H4e are supported. Since both direct and indirect pathways are significant, the flow experience acts as a partial mediator in the mediated pathways of novelty, escape, fascination, and compatibility. The above findings were further confirmed according to Hair and Sarstedt's [71] criteria by removing the indirect effect from the total effect to obtain the VAF (variance accounted for) value. A VAF value less than 20% indicates no mediating effect, a value between 20% and 80% indicates a partial mediating effect and a value greater than 80% indicates a full mediating effect. The results are shown in Table 5, and the VAF values for the four mediated pathways were 59%, 52%, 58%, and 71%, all between 20% and 80%, confirming a partial mediating effect of flow experience.

**Table 5.** Mediator variable analysis.

| Path | Path Coefficient | *t*-Value | Confidence Interval (95%) | | *p*-Value | VAF (%) |
|---|---|---|---|---|---|---|
| | | | Lower Limit | Upper Limit | | |
| NO -> FE -> PTB | 0.379 | 11.99 | 0.318 | 0.44 | 0 | 59 |
| ES -> FE -> PTB | 0.366 | 11.02 | 0.299 | 0.431 | 0 | 52 |
| FA -> FE -> PTB | 0.395 | 11.92 | 0.33 | 0.46 | 0 | 58 |
| CO -> FE -> PTB | 0.492 | 7.386 | 0.358 | 0.619 | 0 | 71 |

Note: NO = Novelty; FE = Flow Experience; PTB = Post-tour Behavior; ES = Escape; FA = Fascination; CO = Compatibility.

**5. Conclusions**

Restorative environmental perception and flow experience reflect the interaction between tourists and tourist places, and tourists' post-tour behaviors reflect the results of this interaction. This study constructs a structural equation model based on SOR theory to clarify the effects of restorative environmental perception and flow experience on the post-tour behaviors of desert off-road self-driving tourists and obtains the following conclusions.

The main dimensions that impact the touring experience of off-road tourists in desert off-road self-driving activities include restorative environmental perception factors like fascination, novelty, escape, and compatibility, as well as flow experience factors like challenge-skill balance, unambiguous feedback, centering of attention, sense of control, loss of self-consciousness, and clear goals. The path analysis in this study successfully validated ten hypotheses. Restorative environmental perception has a direct and positive impact on visitors' flow experiences, which in turn directly and positively influence their post-tour behaviors. Hypotheses H1 and H2 received support. The dimensional analysis of restorative environmental perception revealed that novelty, escape, fascination, and

compatibility have a direct and positive influence on tourists' post-tour behavior. This analysis supported hypotheses H3b, H3c, H3d, and H3e. The analysis of mediating effects indicates that the flow experience partially mediates the relationship between novelty, escape, fascination, compatibility, and tourists' post-tour behaviors. This analysis supported hypotheses H4b, H4c, H4d, and H4e. The research model in this study demonstrates a high overall goodness of fit, indicating strong reliability of the conclusions.

## 6. Discussion

According to the results of the reliability test of the prestudy questionnaire, the coherence dimension of restorative environmental perception and the dimensions of a flow experience of a sense of time distortion, integration of awareness and behavior, and autotelic experience failed the reliability test when used as first-order latent variables of the model. The results suggest that these four dimensions are not the main predictors of tourists' post-tour behaviors in desert off-road self-driving activity and are not part of the internal structural dimensions of the activity. In a study on mountaineering tourism, which also belongs to the adventure tourism category, Wöran and Arnberger [7] confirmed that the coherence dimension had no significant effect on mountaineering activities; Yen and Hsiao [36] confirmed that dimensions such as the sense of time distortion and the integration of awareness and behavior in the flow experience had no significant effect on the emotional state of tourists. The above studies were more similar to the results of our dimensional measures; however, the internal dimensions of restorative environmental perception and flow experience in other settings or activities differed from the findings of this paper. For example, Rosenbaum et al. [19] found that the coherence dimension positively influenced tourists' purchase behaviors while shopping in a mall; the main influential dimension of the flow experience during rafting for tourists was the sense of time distortion [60]. The reason for the dimensional discrepancy is that the activities and sites selected by the researchers differ from those considered in this paper, and related studies have confirmed that the restorative effects of restorative environmental perception vary with environment type [18]; the main dimensions by which the flow experience influences outcomes vary with activity [59]. Therefore, the ten prediction dimensions we constructed are reasonable according to the characteristics of the Kumutag Desert and the specificity of self-driving activities.

As shown by hypotheses H1 and H2 being verified, the desert provides a physical environment that relaxes and challenges people with difficulties, and restorative environmental perception has a direct impact on visitors' flow experiences. Visitors obtain flow experiences through self-driving activities; they not only experience a sense of relaxation away from the world but also challenge their limits and realize their self-worth through off-road self-driving activities. This kind of inner spiritual enjoyment makes tourists' love for the desert increase to awe and attachment, which is the core of tourists' behaviors to repeat and recommend others.

From the results of the validation of hypotheses H3 and H4, the novel, fun, scenic, pristine, and unique desert environment stimulates more interest and enthusiasm in visitors. Tourists can eliminate their worries, empty their minds, do what they like, and achieve physical and psychological recovery in an environment different from those of their daily lives. This pleasant and relaxing feeling allows them to have a satisfying travel experience, which in turn promotes their positive post-tour behaviors. At the same time, the flow experience generated by tourists through overcoming the challenges provided by the environment plays an important mediating role in the relationship between environmental stimuli and tendency behavior. The smooth lines and undulating terrain make the desert mysterious and aesthetically inclined, attracting visitors with a desire to challenge and conquer while viewing the landscape. The flow experience satisfies this experiential purpose of visitors, making it possible for them to enjoy nature not only from external stimuli but also to inspire their souls to resonate with nature, strengthening the motivation and frequency of post-tour behavior.

The high mediating effect of flow experience in the compatible-post-tour behavioral pathway may be due to the following reasons. While dimensions such as fascination, novelty, and escape emphasize the characteristics of the scenic spot itself, compatibility belongs to the deeper dimension of the visitor experience. This dimension focuses on aspects such as whether the tourist place enables the activities that tourists want to participate in and whether the tourist experience meets tourists' expectations. In a study on the restorative environmental perception of tourists in Kanas, the compatibility dimension did not have a significant effect on post-tour behavior [13]. The general excursion and sightseeing class of tourists mainly value the intuitive stimulation brought by the natural elements of the scenic spot and do not feel the charm of the scenic spot through special tourism activities. The characteristics of desert off-road self-driving activities, such as sand dunes with a height difference in dozens of meters in a sea of sand, enable people to experience excitement similar to riding a roller coaster but also the illusion of sailing on the ocean. This experience is unique to the desert environment and is the reason many tourists choose desert off-road self-driving tours. The flow experience generated by this activity deepens the travel experiences of tourists at a psychological level. Therefore, the mediating role of flow experience in the compatible post-tour behavioral path is significantly stronger than those in the other three paths.

## 7. Theoretical Contributions

First, the SOR model is applicable to explain the complex mechanism of action between restorative environmental perception, flow experience, and tourists' post-tour behaviors in desert off-road self-driving activities. The model shows well the internal structure and measurement between the three factors, demonstrates the paths of action among them, and broadens the application of SOR theory in the field of tourism experience.

Second, in previous studies on the influence of restorative environmental perception on tourists' post-tour behaviors, the mediating or involved variables selected were mostly related to place attachment and environmental preferences [75]. Chen et al. [76] used the SOR model as a framework to model the mechanism of influence in the relationships among restorative environmental perception, local attachment, and tourist loyalty, and the results showed a significant mediating role of local attachment between the other two factors. Staats et al. [77] argue that people's restorative needs underlie the generation of environmental preferences, which are implicitly related to tourists' post-tour behavioral preferences as an involved variable. We focused on desert off-road self-driving activities and used flow experience as a mediating variable to construct the research path of restorative environmental perception–flow experience–post-tour behavior. The introduction of the variable of flow experience deepens the mechanism of the influence of restorative environmental perception on tourists' post-tour behaviors, which is important for enhancing adventure tourists' satisfaction and positive post-tour behaviors.

Finally, as a scarce resource with a unique natural landscape, the desert is an excellent place to study restorative environmental perception. However, the research on restorative environmental perception has mainly focused on urban environments and individual tourist attractions, and most of the research subjects have been tourists visiting green environments. We have extended the research perspective to desert adventure tourists, enriching the scope and object of research on restorative environmental perception and contributing to the study of the heterogeneity of restorative environmental perceptions across cultural contexts.

## 8. Management Insights

From Maslow's hierarchy of needs, self-actualization needs are located at the highest level of human needs. Desert off-road self-driving tours can create opportunities for individuals to experience peak feelings and realize their self-worth. The conclusion of this paper can provide a theoretical reference for desert off-road self-driving tourists to realize

high-level spiritual experiences and a new perspective for the environmental development and industry management of the desert off-road self-driving tour market.

From the perspective of restorative environmental perception, desert scenic areas should pay attention to the interactive feelings of visitors and the environment. Fascination reflects that the destination needs to have a beautiful and harmonious natural environment, pleasant climate, etc.; escape reflects that the destination allows tourists to escape from reality and escape from the world. In terms of fascination and escape, the desert attracts tourists with its untouched and pristine appearance and unique mystery, which also means that the desert landscape is more difficult to build and develop, specifically to enhance the cleanliness and coordination of desert environmental resources to continuously attract visitors' attention, to maintain the iconic landscape in the desert, and to set up corresponding textual storytelling to help visitors fully understand the charm of the desert. It allows visitors to fully immerse themselves in the magnificent scenery and forget their daily fatigue and worries. Novelty is used to characterize the destination as having a natural environment that gives people a sense of freshness; compatibility focuses on whether the destination can satisfy the activities that tourists want to do and whether the tourist experience meets the expectations of tourists, focusing on the fit between the scenic spot and the tourists. From the perspective of novelty and compatibility, we should mobilize tourists' longing for desert culture, deepen their participation and integration, and strengthen the excitement and sense of alienation. Construct self-driving tour routes with international popularity, such as tracing the Silk Road, and develop diversified and immersive experience programs to keep tourists coming back. The cultural resources of the cities along the desert are fully explored, displayed and reproduced to make tourists impressed by the trip, not just experience a momentary novelty.

From the perspective of flow experience, driving behavior itself is the reason for the flow experience of tourists, and driving behavior is related to tourists' own skills, which cannot be managed or controlled. However, the special terrain and difficulty of the desert are far beyond the requirements of general driving behavior, so the safety and legality of the industry are a prerequisite to ensure that the activity is safely carried out. Regulating the desert off-road self-driving tour market is a key part of the sustainable and healthy development of adventure tourism and is a necessary requirement to attract tourists seeking high-quality tourism. The government should pay attention to adventure tourism activities and desert off-road self-driving tours to provide the desert vitality, charm, and attractiveness and should introduce relevant policies to encourage and support the self-driving industry. For self-driving leisure travelers, warning signs should be set at difficult nodes, insurance purchase channels should be standardized, and professional medical teams and rescue mechanisms should be established. Professional riders should undergo regular technical evaluation, and chaos, such as private car pickups, should be eliminated.

## 9. Research Limitations and Future Directions

There are several limitations of this study. First, the study area was limited to the Kumutag Desert and did not examine the impact that different desert qualities have on visitors' sense of experience. Follow-up studies should expand the study area and extend the study period to enhance the rationality of the sample and make the findings more explanatory. Second, the physical and mental health status of tourists can impact the flow experience, and this study did not include the variable of tourists' subjective intentions, perhaps limiting the causal link in the findings. Future research on flow experience should be conducted with the help of physiological measurement tools such as ERP and EDA to collect objective physiological index data to reflect the flow experience. Finally, desert off-road self-driving activities require too many driving skills for tourists, and the different driving proficiencies of tourists will cause them to have different travel experiences, which in turn will affect their post-tour behaviors. Future studies could add driver specialization as a moderating variable and set up a control group for the study to compare and

examine whether there is a difference in the flow experience between novice drivers and professional drivers.

**Author Contributions:** Writing—original draft, J.H.; Writing—review and editing, C.L. All authors have read and agreed to the published version of the manuscript.

**Funding:** This research received no external funding.

**Institutional Review Board Statement:** Not applicable.

**Informed Consent Statement:** Not applicable.

**Data Availability Statement:** The data supporting this study's findings are available from the corresponding authors upon reasonable request.

**Conflicts of Interest:** The authors declare no conflict of interest.

## Appendix A

Desert Off-road Self-Driving Tour Questionnaire

Dear Respondent,

This study was conducted to investigate the mechanism of restorative environmental perception on the physical and mental recovery of desert off-road self-driving tourists, to improve the system of desert self-driving tours, and to provide tourists with a better experience of desert self-driving tours. We sincerely invite you to participate in this questionnaire survey, each answer will play a key role in our research, thank you for your participation! Being asked your opinion on some of the questions may cause you some discomfort, but there is no known risk in completing this survey. You may refuse to answer some or all of the questions. If you wish, you may discontinue your participation at any time. Your answers to this questionnaire will be treated as confidential. Please do not disclose any identifying information about yourself in the questionnaire.

Restorative Environmental Perception: With the rapid development of modern society, science and technology continue to progress. The pressure of life and work is also increasingly challenging people's mental capacity. The beautiful environment of tourist destinations can help people relax and reduce fatigue. This kind of environment can help people reduce stress, eliminate all kinds of bad emotions, reduce mental fatigue, and even promote mental and physical health is called "restorative environment". Please complete the following questions, paying attention to the level of differentiation.

**Table A1.** Restorative Environmental Perception Measurement Questionnaire.

| Variables | Items |
| --- | --- |
| Fascination | 1. I want to spend a lot of time exploring the desert.<br>2. The scenery in the desert is charming.<br>3. The scenery in the desert can easily arouse my interest.<br>4. Deserts have attractive qualities. |
| Escape | 1. In the desert I got rid of things I usually have to do.<br>2. The desert gives me a break from the routine of my daily life.<br>3. The desert gives me a sense of freedom from the secular world.<br>4. The desert can help me relax my nervousness. |
| Novelty | 1. I saw many new things in the desert.<br>2. The environment of the desert is much different from my daily life.<br>3. The desert arouses my curiosity.<br>4. The desert brings me a sense of freshness. |
| Compatibility | 1. The desert fits my personality well.<br>2. The things I like to do can be done in the desert.<br>3. I have the feeling of being one with the desert.<br>4. The view of the desert met my expectations.<br>5. I can adapt to the desert environment very quickly. |

Flow Experience: Everyone must have had such a wonderful experience: to devote oneself to something, to enter a state of total concentration, undisturbed, even forgetting the passage of time, only to realize that a long time has passed when it is over, which may be called "entering the state" in Chinese. You must remember the process of hiking or conquering a high mountain. At that time, you were so concentrated that you couldn't feel the passage of time, and you reached a kind of oblivion. Foreign scholars have defined mind flow experience as "mind flow experience is a kind of positive emotional experience accompanied by a high degree of physical and mental commitment to an activity, when people are fully committed to a controllable and challenging activity, they will be immersed in a state of forgetfulness, and when the activity is over or in the middle of the activity when the mind is detached from the activity, they will feel contentment and enjoyment". Please complete the following questions, noting the level of differentiation between the questions.

**Table A2.** Flow Experience Measurement Questionnaire.

| Variables | Items |
| --- | --- |
| Challenge–skill Balance | 1. I can handle the challenges encountered during a desert self-driving tour very well. |
| | 2. My level of driving, exploring, and survival skills is as high as the difficulty of the challenges I encountered during my desert self-driving tour. |
| | 3. I am capable enough to meet the high demands of a desert self-driving tour. |
| Centering of Attention | 1. I can effortlessly focus my attention on desert self-driving tour. |
| | 2. During the desert self-driving tour, I will not think of things that are not related to the tour. |
| | 3. I give my full attention to the desert self-driving tour. |
| Unambiguous Feedback | 1. Based on the feedback on the effectiveness of each activity during the desert self-driving tour, I know I did a good job. |
| | 2. I get timely feedback on the effects of each of my behaviors during the desert self-driving tour. |
| | 3. I can tell that I did a good job from my behavior on the desert self-driving tour. |
| Clear Goals | 1. During the desert self-driving tour, I knew exactly what I was going to do next. |
| | 2. During the desert self-driving tour, I was very clear about what I was operating to achieve. |
| | 3. During the desert self-driving tour, I had a strong desire to accomplish a certain goal. |
| Sense of Control | 1. During the desert self-driving tour, all situations were under my control. |
| | 2. My driving, exploring, and survival skills allow me to do whatever I want on a desert self-driving tour. |
| | 3. I feel in control during a desert self-driving tour. |
| Loss of Self-consciousness | 1. I enjoyed the desert drive so much that I forgot about myself. |
| | 2. I am not worried about my performance during the desert self-driving tour. |
| | 3. I don't care how I behave during the desert self-driving tour. |

**Table A3.** Post-tour Behavior Measurement Questionnaire.

| Variables | Items |
| --- | --- |
| Post-tour Behavior | 1. I am very willing to recommend and share the desert driving activities with others. |
| | 2. I would like to join the desert driving tour again if I have the chance. |

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
