# Peer review of "Restorative Environmental Perception’s Influence on Post-Tour Behavior of Desert Off-Road Self-Driving Tourists: The Mediating Role of Flow Experience"

_sustainability, doi:10.3390/su151712934_

Round 1

Reviewer 1 Report

The article contributes to an insufficiently researched area, namely the mechanisms of the effects of restorative environmental perception and flow experience on the post-tour behaviors of desert off-road self-driving tourists. Theoretical model and research hypothesis are grounded and understandable. The method and results are relevant. The authors discussed the results of the study in the context of expert assessments and international studies, described the theoretical contribution and the possibility of using the research results for decision-making.

I would recommend that the authors change the title of the article, make it shorter, and show the focus of the research in the abstract and keywords.

Author Response

Dear reviewer,

Thank you very much for carefully reviewing this article in your busy schedule, and thank you for your endorsement and advice on the article. Your suggestion for the title of the article is very reasonable, and we are going to change the title of the article to: "Restorative Environmental Perception’s Influence on Post-tour Behavior of Desert off-road Self-driving Tourists:The Mediating Role of Flow Experience"

Best wishes to you.

Reviewer 2 Report

This is an interesting article that focuses on a specific area that does not seem to have been explored much yet. The authors combine interesting concepts such as “restorative environments” and “flow experience” in the desert ecosystem. The authors have bibliographically supported their case and the article has a coherent structure and presentation. Some specific points could be improved. Some relevant suggestions and comments for improvement are proposed.

I read your article with interest since you study an issue that has not been widely explored. There are interesting ideas and combinations of concepts in your study such as restorative environments and flow experience. The choice of the desert as the focus of the research has merit too. From this point of view I think your study is worth being published. The article has a coherent and structured methodology and is comprehensible. I also believe that it is relevant to a wide audience from various scientific fields.

I am listing some points and suggestions for further improvement:

- I think that you have created too many hypotheses for a single article. For example why are separate hypotheses formulated for each one of the  variables of restorative environment perception? What does this choice contribute to the research results?

- It would be more suitable to create a results section separated from the data analysis process. In this way, the research results would be better presented and highlighted.

- It would be also good to create a separate conclusion section.

-The section of “management insights” is not very clear on how it is connected with the rest of the study.

- Generally, it would be helpful to present a clearer scope of the research.

- I noticed that the majority or a large part of the participants of the research are men of middle age and in good economic condition. I wonder if this is a feature of the particular activity or a limitation of the research.

- Finally, I think you have on your hands an excellent and not particularly studied subject - the desert. You have not highlighted the potential of this fascinating ecosystem. You approach the desert environment in an anthropocentric way and refer to it as the background of your research. Desert may not be approached as a background, as an “environmental type” of place, and could not be compared to a shopping mall. The choice not to approach this dynamic ecosystem as a livable place that interacts with its human users deprives the paper of rich results and is inconsistent with contemporary discussions in the field of sustainability. However, probably the orientation of your study is different and I am not asking for it to be overturned.

Author Response

Thank you very much for your busy schedule to read this article carefully, thank you for your recognition of the article and suggestions, I will respond to each of your comments next.

Point 1: I think that you have created too many hypotheses for a single article. For example why are separate hypotheses formulated for each one of the variables of restorative environment perception? What does this choice contribute to the research results?

 Response 1: The central concept of this article is restorative environmental perception, as shown in the literature review, not all the dimensions of restorative environmental perception in previous studies have an impact on tourists' post-tour behavior. This requires this paper to develop a detailed and rigorous study of each dimension, which is one of the innovations of this paper, and I will add the reasons for the dimensional analysis to the literature review as follows, thank you for your comments.

2.5.3. Restorative Environmental Perception and Post-tour Behavior

Restorative environmental perception, which is one of the tourists' motivations to travel, is often associated with post-tour behavior, which is a trip outcome variable. Zhou and Ye used restorative environmental perception as a mediating variable to explore the mechanism of action between tourism involvement and tourists' willingness to revisit and found that restorative environmental perception plays multiple chain mediating roles in the structure of both. Using the context of cultural heritage tourism sites, Korean researchers found that the compatibility dimension had a positive effect on tourists' revisit intentions. In a study of tourists' perceptions of restorative environments in Guangdong's Nankunshan tourist resort, Chen and Xi found that the fascination and compatibility dimensions significantly and positively influence tourists' post-tour behavioral intentions. In a study on the restorative environmental perception of tourists in Kanas, Huang et al. found that the novelty and escape dimensions significantly and positively influence post-tour behavior. Environmental psychology suggests that the restorative perceptions of visitors vary with the type of environment and that there is not a parallel structure between restorative environmental perception dimensions but a differentiated and structural one. Previous literature suggests that not all dimensions impact post-tour behavior, thus necessitating individual and meticulous examination of each dimension to refine the effects of restorative environmental perception on tourists’ behavior in desert off-roading scenarios. This will also allow exploration of the dimensions that hold significance in these scenarios. Deserts possess restorative elements such as vast spatial structures and novel natural scenery, which are more immersive to tourists than traditional scenic spots, so the unique desert environment may have a different impact on tourists' post-tour behaviors. During the self-driving process, tourists temporarily get away from worldly worries, integrate themselves into the environment and gain a restorative experience. This experience can make tourists deepen their favorable feelings for and identification with the destination, which in turn can influence their post-tour behaviors. Therefore, we propose the following hypothesis.

Point 2: It would be more suitable to create a results section separated from the data analysis process. In this way, the research results would be better presented and highlighted.

Point 3: It would be also good to create a separate conclusion section.

 Response 2、3: Your suggestions are very reasonable and I will include a separate section on the findings of the study in Part V. Thank you for your suggestions.

5 Conclusions

Restorative environmental perception and flow experience reflect the interaction between tourists and tourist places, and tourists' post-tour behaviors reflect the results of this interaction. This study constructs a structural equation model based on SOR theory to clarify the effects of restorative environmental perception and flow experience on the post-tour behaviors of desert off-road self-driving tourists and obtains the following conclusions.

The main dimensions that impact the touring experience of off-road tourists in desert off-road self-driving activities include restorative environmental perception factors like fascination, novelty, escape, and compatibility, as well as flow experience factors like challenge-skill balance, unambiguous feedback, centering of attention, sense of control, loss of self-consciousness, and clear goals. The path analysis in this study successfully validated ten hypotheses. Restorative environmental perception has a direct and positive impact on visitors’ flow experiences, which in turn directly and positively influence their post-tour behaviors. Hypotheses H1 and H2 received support. The dimensional analysis of restorative environmental perception revealed that novelty, escape, fascination, and compatibility have a direct and positive influence on tourists’ post-tour behavior. This analysis supported hypotheses H3b, H3c, H3d, and H3e. The analysis of mediating effects indicates that the flow experience partially mediates the relationship between novelty, escape, fascination, compatibility, and tourists’ post-tour behaviors. This analysis supported hypotheses H4b, H4c, H4d, and H4e. The research model in this study demonstrates a high overall goodness of fit, indicating strong reliability of the conclusions.

Point 4: The section of “management insights” is not very clear on how it is connected with the rest of the study.

Response 4: The Management Insights section is centered on the desert off-road self-driving tour program, which aims to improve the quality of off-road tourists' tour. Based on the article's conclusions we can learn the characteristics of the dimensions that influence post-tour behavior and propose appropriate management solutions in terms of both restorative environmental perception and flow experiences. This is where the findings contribute to the Management Insights, making it easier for managers to identify the relevant dimensions and make adjustments to the desert off-road program based on their importance. I will re-emphasize the dimensional characteristics of restorative environmental perception and flow experiences in the Management Insights section in order to deepen the connection with the previous paper, as shown below, and thank you very much for your comments.

8 Management Insights

From Maslow's hierarchy of needs, self-actualization needs are located at the highest level of human needs. Desert off-road self-driving tours can create opportunities for individuals to experience peak feelings and realize their self-worth. The conclusion of this paper can provide a theoretical reference for desert off-road self-driving tourists to realize high-level spiritual experiences and a new perspective for the environmental development and industry management of the desert off-road self-driving tour market.

From the perspective of restorative environmental perception, desert scenic areas should pay attention to the interactive feelings of visitors and the environment. Fascination reflects that the destination needs to have a beautiful and harmonious natural environment, pleasant climate, etc.; escape reflects that the destination allows tourists to escape from reality, escape from the world. In terms of fascination and escape, the desert attracts tourists with its untouched and pristine appearance and unique mystery, which also means that the desert landscape is more difficult to build and develop, specifically to enhance the cleanliness and coordination of desert environmental resources to continuously attract visitors' attention, to maintain the iconic landscape in the desert, and to set up corresponding textual storytelling to help visitors fully understand the charm of the desert. It allows visitors to fully immerse themselves in the magnificent scenery and forget their daily fatigue and worries. Novelty is used to characterize the destination as having a natural environment that gives people a sense of freshness; compatibility focuses on whether the destination can satisfy the activities that tourists want to do, and whether the tourist experience meets the expectations of tourists, focusing on the fit between the scenic spot and the tourists. From the perspective of novelty and compatibility, we should mobilize tourists' longing for desert culture, deepen their participation and integration, and strengthen the excitement and sense of alienation. Construct self-driving tour routes with international popularity, such as tracing the Silk Road, and develop diversified and immersive experience programs to keep tourists coming back. The cultural resources of the cities along the desert are fully explored, displayed and reproduced to make tourists impressed by the trip, not just experience a momentary novelty.

From the perspective of flow experience, the specificity of the location and technology of desert off-road activities is the main reason for tourists' flow experience. The dimensions of centering of attention, sense of control, and challenge-skill balance, which have an impact on tourists' post-tour behavior, are generated by driving behavior, which is related to tourists' own skills and cannot be managed or controlled. However, the special terrain and difficulty of the desert is far beyond the requirements of general driving behavior, so the safety and legality of the industry is a prerequisite to ensure that the activity is safely carried out. Regulating the desert off-road self-driving tour market is a key part of the sustainable and healthy development of adventure tourism and is a necessary requirement to attract tourists seeking high-quality tourism. The government should pay attention to adventure tourism activities and desert off-road self-driving tours to give the desert vitality, charm and attractiveness and should introduce relevant policies to encourage and support the self-driving industry. For self-driving leisure travelers, warning signs should be set at difficult nodes, insurance purchase channels should be standardized, and professional medical teams and rescue mechanisms should be established. Professional riders should undergo regular technical evaluation, and chaos, such as private car pickups, should be eliminated.

Point 5: Generally, it would be helpful to present a clearer scope of the research.

Response 5: The study area of this paper is the Kumutag Desert, and the questionnaire distribution site, the Dragon Inn off-road base, is located at the northern edge of the Kumutag Desert, which is one of the favorite starting points for desert off-road self-driving tourists. The status and prestige of Dragon Inn in the field of desert off-road has been mentioned at the preface, and I will introduce the base in detail again at the overview of the study area as follows, and your comments will be greatly appreciated.

3.2. Overview of The Study Area

The Kumutag Desert is located at the southern end of the old city of Shanshan County, Xinjiang, and is ranked as one of the eight deserts of China, with a total area of 2,500 square kilometers. As the closest desert the city, it has high accessibility and strong transportation convenience and is close to tourist cities such as Urumqi, Hami and Korla, making its geographical location unique. This desert is in the temperate continental climate zone, with temperature suitable for year-round adventure and recreation, sand sports, sand therapy, health care, sightseeing and other activities. In 2015, the East Lake Auto Self-Driving Camp was built in the desert hinterland, providing a leisure place and living convenience for desert off-road self-driving tourists, explorers and travelers. Compared with other deserts in China, the Kumutag Desert has a high water content, fine and pure sand, and sand dunes with undulating drops of up to 100 meters. The complete types of wind and sand landscapes and the peculiar landscape are conducive to the determination of the restorative environmental perceptions of tourists. In the first new mode of tourism combining city and desert, it creates a sense of atmosphere far from the world and is vast and primitive, becoming a competitive paradise and beloved place for desert off-road self-driving enthusiasts.

The survey was distributed at the Dragon Inn off-road base, situated on the northern edge of the Kumutage Desert, spanning an area of 0.02 square kilometers. Proximity to the Kumutag Desert highway ensures convenient access for tourists. Additionally, a visitor service center is available, offering complimentary car and water refueling services. This base is a renowned attraction for self-driving enthusiasts who engage in desert off-road adventures. It serves as the starting point for desert off-road leaders such as the Bohai Off-road Club and the Avenue Off-road Club, and serves as a cultural hub for off-road activities in Xinjiang. The research subjects here differ significantly from general tourists.

Point 6: I noticed that the majority or a large part of the participants of the research are men of middle age and in good economic condition. I wonder if this is a feature of the particular activity or a limitation of the research.

 Response 6: You have observed the subjects of this paper very carefully, and in my research I have also found that the desert off-road activity crowd is characterized by a preponderance of middle-aged men in good financial standing, which is both a characteristic and a limitation of the activity. I also raised this question when I consulted with the president of the off-road club and they gave the following response. Desert off-road driving tour is divided into two categories, one belongs to the mass tourism, applicable to all ages, tourists can spend a hundred dollars in the desert scenery by professionals driving the vehicle to feel the desert off-road. The other is a niche high-end tours, by the owner himself to drive modified vehicles into the desert, which requires owners to have good economic conditions and physical quality. The mediating variable in this paper is the flow experience, which can be conceptualized as requiring the person to personally operate the practice to get the feeling. From the conclusion of the article, it can be seen that the dimensions of challenge and skill balance, sense of control, and timely feedback are all felt by the owner in the driving activity, and for the general public tourists, who are unable to have a driving experience, the felt flow experience is relatively weak, which determines that the object of this paper is the niche high-end off-road tourists, in order to ensure that the research object's flow experience is more rich and profound. This is the innovation of this paper, but of course it also creates some limitations, so this paper suggests that future research could focus on how the restorative environmental perception and flow experiences of desert off-road tourists in mass tourism differ from this paper.

Point 7: Finally, I think you have on your hands an excellent and not particularly studied subject - the desert. You have not highlighted the potential of this fascinating ecosystem. You approach the desert environment in an anthropocentric way and refer to it as the background of your research. Desert may not be approached as a background, as an “environmental type” of place, and could not be compared to a shopping mall. The choice not to approach this dynamic ecosystem as a livable place that interacts with its human users deprives the paper of rich results and is inconsistent with contemporary discussions in the field of sustainability. However, probably the orientation of your study is different and I am not asking for it to be overturned.

 Response 7: I am inspired by your very unique insights. The research concept of this paper is centered around off-road driving activities in the desert, the desert is the place where off-road activities occur in the text, and the description of the desert focuses on how its topographical features support off-road activities to take place, which may have resulted in a more homogenous and rigid focus on the desert in this paper. Your idea of studying the desert as a dynamic ecosystem is very much appreciated, and I will follow up by continuing to focus on the restorative characteristics that the desert itself possesses, with an emphasis on the impacts that the desert's own qualities have on people, rather than focusing on a particular desert activity. Thank you very much for your suggestions.

Reviewer 3 Report

Dear Authors,

This is a well-written paper and it is a topic that is highly under-researched. The methods are sound and the implications and future research are appropriate.

I am recommending this paper for publication however, be aware that it is so niche citations in future research will be few.

Best of luck,

Author Response

Dear reviewer,

Thank you very much for carefully reviewing this article in your busy schedule, and thank you for your endorsement and advice on the article. This article does belong to a niche research area, but this area has great research potential, and we will continue to pay attention to the content related to desert restorative environmental perception and conduct in-depth research.

Best wishes to you.